# Parasitic Contamination of Fresh Leafy Green Vegetables Sold in Northern Lebanon

**DOI:** 10.3390/pathogens12081014

**Published:** 2023-08-04

**Authors:** Dima El Safadi, Marwan Osman, Angel Hanna, Iman Hajar, Issmat I. Kassem, Sara Khalife, Fouad Dabboussi, Monzer Hamze

**Affiliations:** 1Department of Clinical Sciences, Liverpool School of Tropical Medicine, Liverpool L7 8XZ, UK; dima.elsafadi@lstmed.ac.uk; 2Laboratoire Microbiology, Santé et Environnement, Doctoral School of Sciences and Technology, Faculty of Public Health, Lebanese University, Tripoli 1300, Lebanon; angellhanna@outlook.com (A.H.); imanhajjar56@gmail.com (I.H.); fdaboussi@hotmail.com (F.D.); mhamze@monzerhamze.com (M.H.); 3Cornell Atkinson Center for Sustainability, Cornell University, Ithaca, NY 14853, USA; 4Department of Public and Ecosystem Health, College of Veterinary Medicine, Cornell University, Ithaca, NY 14853, USA; 5Center for Food Safety, Department of Food Science and Technology, University of Georgia, Griffin, GA 30223, USA; issmat.kassem@uga.edu; 6Department of Medical Laboratory Technology, Faculty of Health Sciences, Beirut Arab University, Tripoli 1300, Lebanon; sara.khalifeh@bau.edu.lb

**Keywords:** parasites, *Blastocystis* spp., *Ascaris* spp., food safety, leafy green vegetables, fresh produce, risk factors, public health, Lebanon

## Abstract

Contaminated, raw or undercooked vegetables can transmit parasitic infections. Here, we investigated parasitic contamination of leafy green vegetables sold in local markets in the Tripoli district, Lebanon, during two consecutive autumn seasons (2020–2021). The study involved the microscopic examination of 300 samples of five different types of vegetables (60 samples per type) and used standardized qualitative parasitological techniques for some protozoa and helminths. The results showed that 16.7% (95% interval for p: 12.6%, 21.4%) (50/300) of the vegetable samples were contaminated with at least one parasite. The most frequently detected parasite was *Blastocystis* spp. (8.7%; 26/300); this was followed in frequency by *Ascaris* spp. (3.7%; 11/300). Among the different vegetable types, lettuce (23.3%; 14/60) was the most contaminated, while arugula was the least contaminated (11.7%; 7/60). The statistical analysis did not reveal any significant association between the prevalence of parasitic contamination and the investigated risk factors, which included collection date, vegetable type, market storage status, and wetness of vegetables at the time of purchase (*p* > 0.05). The high prevalence of parasitic contamination also suggested the potential presence of other microbial pathogens. These findings are important because leafy green vegetables are preferentially and heavily consumed raw in Lebanon. Thus, implementing effective measures that target the farm-to-fork continuum is recommended in order to reduce the spread of intestinal pathogens.

## 1. Introduction

Vegetables are a fundamental component of the Mediterranean diet, offering essential nutrients and promoting satiety [1]. However, consuming contaminated vegetables poses a significant food safety risk, leading to serious infections. While contaminated leafy greens have been associated with numerous foodborne outbreaks across the globe yearly, they also represent the leading cause of foodborne illnesses in the USA [2]. Between 2014–2021, the CDC recorded 78 foodborne disease outbreaks associated with the consumption of leafy greens, mainly lettuce. These resulted in 2028 illnesses, 477 hospitalizations, and 18 deaths; however, these numbers are considered to be underestimations because foodborne illnesses are usually underreported [3]. While most foodborne illnesses are typically mild and self-limiting, low- and middle-income countries and vulnerable populations are affected more severely, experiencing comparatively higher morbidity and mortality rates associated with foodborne infections [4].

Vegetables are susceptible to contamination by human pathogens at any point of the farm-to-fork continuum, including at pre- and postharvest [5,6]. At preharvest, water, manure (of human or animal origin), and soil represent the primary sources of contamination [7]. For example, the use of contaminated water for irrigation, the utilization of improperly composted/untreated manure as fertilizer, or prior land use and agricultural practices can substantially elevate microbial loads on crops and affect produce safety [7,8], as also shown in Lebanon [8]. Microbial pathogens can endure in the matrices (water, manure, and soil) for prolonged periods, depending on matrix type, the specific pathogen, agricultural practices in use, and initial microbial load [5]. At postharvest, improper washing, handling, and storage among other issues have been implicated in produce contamination and associated outbreaks [9]. These observations were confirmed by the analysis of 167 multistate outbreaks that occurred between 2006 and 2021 in the USA. The etiologic agents associated with these outbreaks were confirmed, and the source was largely attributed to cross-contamination within the distribution chain, poor agricultural practices, and fresh importations [10].

Lebanon’s Mediterranean climate, arable land, and relatively abundant water resources are suitable for the cultivation of fresh vegetables, which are consumed regularly in the country [11]. Notably, the average per capita consumption of vegetables was reported to be 185 kg annually during 2003–2005 [12]. However, Lebanon has been facing well-documented challenges in food safety, pollution (water quality, solid waste management), and its economy [13,14,15,16,17]. Indeed, despite the abundance of fresh produce, several investigations have argued that undesirable farming practices, which entail using untreated wastewater and other contaminated water sources for irrigation [8,11,18], pose a substantial contamination risk and jeopardize the safety of locally grown vegetables. Additionally, recent research has identified elevated densities of fecal coliforms and multi-drug resistant *Escherichia coli* in river water and other surface water reservoirs used for irrigation across the country [14,19]. Specifically, based on the densities of fecal indicators, up to 74% of water samples collected from major rivers in Lebanon exceeded the microbiological acceptability standards for irrigation. This is notable, because river water is the major source for the irrigation of crops in the country. Currently, Lebanon has a very limited capacity for wastewater management, a fact that is resulting in widespread water pollution. The latter issue, along with the chronic water resource mismanagement that results in freshwater scarcity, has forced farmers to routinely rely on untreated and contaminated water sources [20,21,22]. Subsequently, fresh leafy vegetables are at elevated risk of contamination, posing a real concern to stakeholders in Lebanon. Agricultural fields in Lebanon are located in some of the most deprived areas in the country. It is known that foodborne and waterborne diseases disproportionally affect disenfranchised populations that do not have access to clean water and adequate sanitation. Indeed, we have previously documented concerning trends in food safety and gastrointestinal diseases in these populations in Lebanon [15,23]. Taken together, these observations suggest that contaminated fresh vegetables pose a risk that must not be overlooked in Lebanon.

Studies on the contamination of fresh vegetables in Lebanon are very limited and mainly focus on bacterial contamination or indicators. Despite the endemicity of intestinal parasitosis in Lebanon [24,25,26], there are no available data on the extent of parasitic contamination in vegetables sold on Lebanese markets. Previously, we showed that *Cryptosporidium* spp. and *Giardia duodenalis* were among the most frequently detected intestinal parasites in disenfranchised outpatients in North Lebanon suffering from acute diarrhea [27]. These parasites were also previously reported in high percentages in children in North Lebanon [25]. Consequently, this study aimed to assess the prevalence of parasitic contamination and associated risk factors during the marketing phase of frequently consumed vegetables in North Lebanon. The ultimate objective is to provide valuable insights that can be utilized in future research or in combination with other findings to inform public health and implement agricultural policies and practices.

## 2. Materials and Methods

### 2.1. Study Design and Area

This pilot cross-sectional study was conducted in the Tripoli district of the North Governorate in Lebanon, the country’s second-largest city and the primary urban center in northern Lebanon. With a population of approximately 350,000 inhabitants, the city is located at 34°26′12″ N latitude and 35°50′58″ E longitude and experiences moderately hot summers and mild winters. Considering that warm seasons are associated with a higher prevalence of parasitic contamination in vegetables [28], the study focused on collecting fresh vegetable samples during the autumn months, characterized by relatively hot temperatures ranging from 29 °C to 14 °C, respectively, and occasional rainfall. This period is characterized by relatively hot temperatures ranging from 29 °C to 14 °C and occasional rainfall. The local market selection was conducted using a random sampling approach in order to achieve a representative distribution across the Tripoli district, as depicted in Figure 1. To assess the microbial risk inherent in Lebanese traditional cuisine, five leafy green vegetables that are frequently consumed as raw materials were purposefully chosen, including mint, lettuce, parsley, arugula, and purslane.

### 2.2. Sample Collection

A total of 300 fresh vegetable samples were collected from 68 local markets in Tripoli over two consecutive years (2020–2021), with the annual collection of 150 samples. Fifteen samples of vegetable samples were collected weekly over a period of 10 weeks (from 1st of September until mid-November). Sixty samples were randomly selected for each of mint (*Mentha piperita*), lettuce (*Lactuca sativa*), parsley (*Petroselinum crispum*), arugula (*Eruca vesicaria*), and purslane (*Lepidium sativum*). A single batch/bundle of a given vegetable sample was purchased once from each seller and was collected in a separate sterile and labelled plastic bag. The samples were transported to the microbiology laboratory in a cooler with ice (4 °C) on the day of purchase and processed immediately.

The vegetables were sourced from different farms and agricultural areas located in both the Bekaa Valley, a region in Eastern Lebanon that accounts for approximately 37% of the country’s vegetable production [29], and North Lebanon. Prior to sample collection, oral consent was obtained from the sellers. Pertinent retail parameters were meticulously recorded, including the type of market construction (closed/open), the mode of vegetable display (on shelves/on the ground/in a box), and the moisture of the vegetable at the time of purchase (wet/dry). The open-air market is defined as an outdoor market that takes place in an open public space. The closed market is an indoor market that operates within an enclosed or covered space. All the above factors were investigated as potential risk factors for parasitic contamination in the retail settings.

### 2.3. Sample Processing

The edible portions of the vegetable samples were separated from the non-edible parts in accordance with household practices. Subsequently, 30–40 g of fresh vegetable samples was washed with 75 mL of 10% formalin saline, as previously described [30]. The samples were homogenized at a low speed for 1 min in a Stomacher^®^ 80 Biomaster to facilitate the detachment of any parasite stages such as oocysts, cysts, eggs, or larvae. The washing solution was then filtered through gauze and centrifuged at 3000 rpm for 5 min in order to concentrate the parasite stages [31]. The sediment was gently agitated and then subjected to qualitative examination under a light microscope using saline and iodine wet mount identification techniques. The identification of parasites was based on the morphological characteristics of cysts, eggs, or larvae [32].

### 2.4. Statistical Analysis

Descriptive and statistical analysis was performed using the R software (R Core team, version 4.1.0; R Studio, version 1.4.1106) with several packages (e.g., dplyr, tidyr, summarytools). The results were plotted using the ggplot2 R package. Categorical data were described as frequencies and associated percentages. The difference between the proportions of intestinal parasites among different categories was initially compared using the chi-squared test. Subsequently, a multivariable logistic regression model was built with parasitic contamination of the vegetables as the outcome and the date of collection, sample type, market storing status, and wetness of vegetables as predictors. The statistical tests were two-sided, with a type I error set at α = 0.05. The code necessary to replicate the analysis is publicly available (DOI: 10.5281/zenodo.7576370).

## 3. Results

The majority of markets (62.7%; 188/300) were observed to be of an open-air type, while 37.3% (112/300) were enclosed. A large percentage of the samples (44.7%; 134/300) was found to be wet when purchased, implying exposure to water. The vegetable samples were mainly displayed on shelves (90.3%; 271/300), with a small percentage being kept on the ground (1.6%; 5/300) or in boxes (8%; 24/300). Our results showed that 50 vegetable samples (16.7%; 95% interval for p: 12.6%, 21.4%) were positive for at least one parasite. As shown in Table 1, lettuce was the most contaminated sample type, with 23.3% (14/60) of the lettuce samples being positive, which constituted 28% (14/50) of the overall positive vegetable samples. Arugula was the least contaminated sample type, with an 11.7% (7/60) rate.

Most samples were contaminated with only one parasite (94.0%; 47/50). Polyparasitism was detected in three samples (6%; 3/50), where *Entamoeba* spp./*Ascaris* spp. and *Entamoeba* spp./*Blastocystis* spp. were detected in individual samples of mint and arugula, respectively. Three parasites, namely, *Blastocystis* spp., *Ascaris* spp., and *Toxocara* spp., were reported in one lettuce sample. *Blastocystis* spp. and *Ascaris* spp. were the most frequently detected parasites in the vegetables, being found in of 8.7% and 3.7% of samples, respectively (Figure 2). Importantly, the distribution of the intestinal parasites varied between the selected fresh vegetable types. For example, *Blastocystis* spp. was the most common parasite detected in lettuce and purslane, while *Ascaris* spp. and *Entamoeba* spp. were found in all vegetable types except arugula and lettuce, respectively. Hookworm and *Giardia* spp. were both detected in mint and arugula.

The univariate and multivariable statistical analyses showed no significant association between parasitic contamination and sample type. Likewise, there was no such link with the other variables, including the collection date, market storage status, and moisture status of vegetables when purchased (*p*-value > 0.05) (Table 1).

## 4. Discussion

Consuming contaminated raw vegetables can have serious implications for public health because they can be a potential source for the transmission of parasitic foodborne diseases [33]. Among different vegetable groups, leafy green vegetables are of particular concern due to their high susceptibility to contamination. This study has revealed that a range of pathogenic protozoa and helminths, both of human and other zoonotic origins, are present on vegetable samples collected from Lebanese markets, including *Blastocystis* spp., *Entamoeba* spp., *Balantidium* spp., and *Giardia* spp., as well as soil-transmitted helminths such as *Ascaris* spp., *Trichuris* spp., *Strongyloides* spp., *Toxocara* spp., and hookworms. To our knowledge, this is the first study to evaluate the parasitic contamination of leafy greens being retailed in Lebanon.

Our results highlighted a relatively high prevalence of parasitic contamination in vegetables sold in North Lebanon. This aligned with a previous study reporting a prevalence of 16.67% in vegetables directly harvested from the Bekaa Valley in 2020 [34]. The previous study suggested that the water used for irrigation, sourced from the Litani river, had significant levels of contamination with *Escherichia coli* and some parasite species in comparison to groundwater and treated wastewater, surpassing the threshold limits recommended by the FAO [35]. Moreover, crops irrigated with this water showed the highest percentage of parasitic contamination when compared to those irrigated with groundwater and treated wastewater. Notably, *Ascaris lumbricoides* and *Blastocystis* spp. were the most frequently detected parasites, and were predominantly found in radish samples [34]. Furthermore, lettuce and parsley samples grown in the Bekaa Valley were found to be contaminated with a considerable load of bacterial pathogens, including *Salmonella* spp. and *Staphylococcus aureus* [8,34,36]. Importantly, the Kadisha-Abou Ali River situated in North Lebanon has been deemed one of the most heavily contaminated rivers in Lebanon, primarily due to human activities [37,38]. The water quality of this river has been reported to be at its worst in the Tripoli district, with pollution accumulation from the upstream flow exacerbated by local wastewater discharge [39]. Additionally, inadequate waste management infrastructure, indiscriminate waste disposal in rivers due to urbanization [40], and pollution from makeshift refugee camps and settlements on the banks of the rivers [21] have exacerbated the use of contaminated water for irrigation. Although water is a significant factor contributing to the microbial contamination of vegetables, other potential causes related to the inadequate implementation of good post-harvest and agricultural practices have been observed. These include the lack of personal hygiene among farmworkers, the unviability of proper hygienic resources during harvest, and the limited awareness to GAPs among others [8,41].

When compared to other countries that use microscopic examination as a detection method, the percentage of contaminated vegetables in Lebanon was lower than Syria (34.4%) [42], Egypt (39%) [43], and Iran (60.3%) [44]. However, it was higher than the percentage reported in Turkey (2.4%) [45] and comparable to the levels found in Saudi Arabia (16%) [46] and the United Arab Emirates (15.1%) [47]. The variability in the contamination between countries could be attributed to several factors, including environmental and climatic conditions, farming and post-harvest handling practices, sanitation standards, and differences in study design. This final category ranges from the choice of vegetable varieties to the elution solution and the number of samples analyzed.

*Blastocystis* spp. was the most detected parasite in the vegetables. This is likely because of the diverse routes involved in contaminating vegetables, including sewage, tap water, well water, and untreated river water [48]. This parasite has also been found in high percentages in dairy cattle (63.4%) and poultry (32%) raised in the north of Lebanon [49,50]. As a result, using chicken or livestock manure as fertilizer in crop production might increase the risk of zoonotic parasitic contaminants being on vegetables, including *Blastocystis* spp. Furthermore, *Entamoeba* spp. had a low prevalence in our samples; however, the pathogenic parasite *E. histolytica* can be transmitted easily in outbreak settings due to the low infectious dose of the cysts and their relative resistance to chlorine [51]. *Ascaris* spp. is another helminth commonly associated with the use of wastewater for irrigation and human excreta for crop fertilization. The eggs of this parasite are highly resistant in the external environment and can remain viable for up to 15 months in surface water [52], which contributes to its effective dissemination. The World Health Organization has stated that the abundance of helminth eggs during hot weather is due to certain environmental conditions, such as humidity, temperature, and sunlight, that facilitate their maturation [53].

The main parasites detected in the present study are consistent with those found in the Lebanese population. *Blastocystis* spp. was found to be the most prevalent enteric parasite in Northern Lebanon, with prevalences of 19.6% in the general population [24] and 63% in schoolchildren [25]. However, *Giardia duodenales* was detected among 25.8% of schoolchildren in Tripoli in 2016 and only 1% of the analyzed vegetable samples were positive. *Giardia* cysts were found to exhibit a limited ability to survive on fresh produce for an extended period as 50% of the cysts died within the first 24 h [54]. Regarding roundworms, the prevalence of *Toxocara* spp. and *A. lumbricoides* in the Lebanese population was 19% in 2006 [55] and less than 1% in 2016 [25], respectively. In contrast, hookworms, *Strongyloides* spp., and *Trichuris* spp. were rarely detected in Lebanon, with a prevalence not exceeding 0.2% for *Trichuris trichiura*, 0.1% for hookworm, and 0.01% for *Strongyloides stercolaris* [56].

In this study, lettuce was found to be the most contaminated vegetable type, as in many other studies carried out in Arab countries [42,57]. Several factors related to the vegetable type could influence the level of parasitic contamination, such as texture, size, uneven surfaces, distance from the soil, and the flexibility of the leaves. Regardless, the detection of parasites contaminating vegetables is a helpful indicator of the incidences of enteric bacterial pathogens in the community. Interestingly, a recent cholera outbreak was declared in Lebanon, resulting in 6158 suspected and confirmed cases and 23 associated deaths as of 17th of January 2023 [58]. The spread of the disease was also facilitated by the deficiency in wastewater treatment systems and the collapse of sanitation and infrastructure systems due to an unfolding economic crisis [21]. Consequently, if the economic situation and water pollution in the country become worse, we predict the emergence and further occurrence and transmission of a plethora enteric pathogens, including foodborne parasites.

The retail market factors examined in this study did not show a significant association with parasitic contamination. This aligned with findings from prior research where variables such as vegetable display methods [59,60], vegetable types [61] and market storing status [59] were also found to be non-significant in their impact. In addition, the wetness status of the vegetables analyzed in this study are attributed to practices such as surface-sprinkling with water. A study in Sudan revealed a high contamination of the water used to sprinkle vegetables with *Strongyloides* larvae, *Giardia duodenalis*, *Entamoeba coli*, and *Ascaris lumbricoides* [30]. The lack of significance in our study may be attributed to the low number of wet vegetable samples. In this context, previous investigations in Turkey, Iran and Ethiopia showed that washing of vegetables contributes to a significant decrease in the rate of parasitic contamination [59,62,63]. 

This study had some limitations that must be considered. First, the number of collected vegetable samples was relatively low due to the COVID-19 pandemic national restrictions, which may have affected the representativeness of the findings. Additionally, the use of microscopic examination as a detection method also has limitations in terms of sensitivity compared to other techniques, which are not widely available in Lebanon. Recently, the FLOTAC technique has shown an additional asset by detecting gastrointestinal parasites in vegetables in Brazil [64]. Molecular investigations, as shown by Barlaam et al. (2022) in their recent study conducted in Italy, have represented a valuable advancement compared to traditional microscopy, particularly in the analysis of fresh produce [65]. Therefore, support for the optimization and implementation of molecular techniques, which can detect all the parasite species present in vegetable samples, is critical for future surveillance studies in Lebanon.

## 5. Conclusions

This pilot study provides valuable insights into the prevalence of the parasitic contamination of fresh vegetables sold in Lebanese markets. Data on foodborne and waterborne parasites in Lebanon are limited, especially in food and water matrices. Our results highlighted the important role of produce as a potential source of parasitic infections in the community. The high prevalence of contamination underscores the need for effective measures to prevent the transmission of enteric pathogens via contaminated irrigation water and vegetables. It is recommended that stakeholders implement a proper sanitation system, monitor the application of GAPs, GHPs, and HACCP systems in the food sector, periodically screen on-market vegetables, and raise awareness among consumers about the importance of proper vegetable washing, decontamination, and consumption. Further studies with larger sample sizes and more comprehensive analysis of parasitic contamination in vegetables in different regions of Lebanon are warranted.

## Figures and Tables

**Figure 1 pathogens-12-01014-f001:**
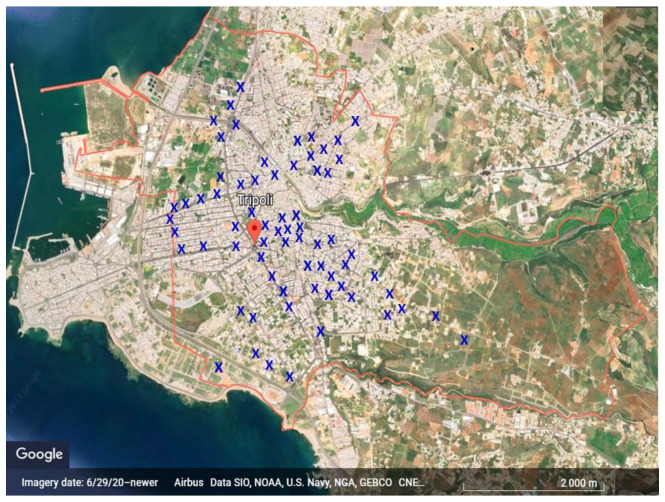
Map of sampling and study location (adapted from Google Earth, 2023). Blue X represents the locations of the 68 local markets in Tripoli where fresh vegetable samples were collected.

**Figure 2 pathogens-12-01014-f002:**
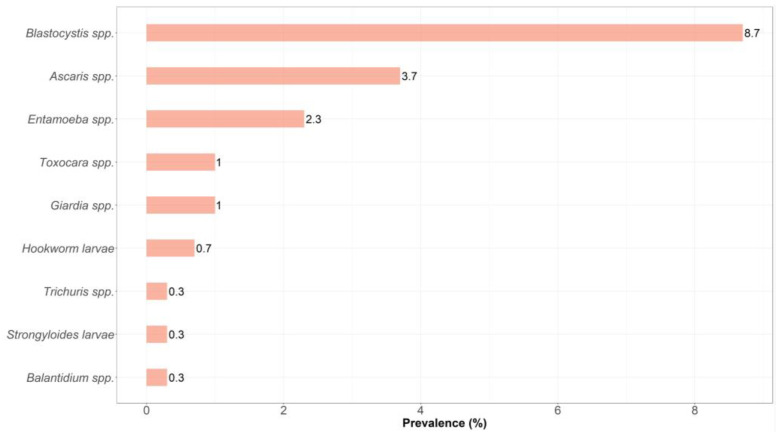
Prevalence of intestinal parasites in commonly consumed local raw vegetables sold in the Lebanese markets in North Lebanon.

**Table 1 pathogens-12-01014-t001:** Determinants of intestinal parasitic contamination in commonly consumed local raw vegetables sold in the Lebanese market in Northern Lebanon.

	Number of Vegetable Samples	Presence of Parasite (%)	Univariate Analysis *X*^2^ (*p*-Value)	Multivariable Analysis OR (95% CI)
Date				
2020 ^1^	150	29 (19.3%)	1.18 (0.28)	
2021	150	21 (14.0%)		0.77 (0.25–2.28)
Sample type				
Arugula ^1^	60	7 (11.7%)	3.36 (0.50)	
Lettuce	60	14 (23.3%)		2.36 (0.90–6.74)
Mint	60	11 (18.3%)		1.72 (0.62–5.02)
Parsley	60	9 (15.0%)		1.35 (0.47–4.05)
Purslane	60	9 (15.0%)		1.37 (0.47–4.14)
Market storing status				
Closed ^1^	112	16 (14.3%)	0.48 (0.49)	
Open	188	34 (18.1%)		0.97 (0.37–2.44)
Purchase status				
Dry ^1^	166	24 (14.5%)	0.97 (0.32)	
Wet	134	26 (19.4%)		1.23 (0.51–3.04)

^1^ Reference group.

## Data Availability

The original collected data and the code necessary to replicate the statistical analysis are publicly available (DOI: 10.5281/zenodo.7576370).

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
