# Peer review of "Parasitic Contamination of Fresh Leafy Green Vegetables Sold in Northern Lebanon"

_pathogens, 2023, doi:10.3390/pathogens12081014_

Round 1
Author Response
Dear Reviewer 1,
We are pleased that you found our manuscript interesting, and we thank you for the thoughtful reading and constructive comments.
Please find a revised version of our manuscript. As requested, we answered all your comments and suggestions. All answers are listed below and included in the revised manuscript.
Thank you for considering this revised version of our manuscript.
Comment 1. The introduction is well organized but rather redundant and should include updated references. The introduction should be shortened. Line 49: A reference is needed to support the sentence "Between 2014-2021 (...) underreported".
REPLY: As requested, we have shortened the introduction section and have added a reference to the mentioned expression “Between 2014–2021 (…) underreported (3).
CDC. Lettuce, Other Leafy Greens, and Food Safety 2023 [Available from: https://www.cdc.gov/foodsafety/communication/leafy-greens.html.
Comment 2. Line 54 Ref 4: The cited reference is too old. Authors are invited to refer to more recent contributions (e.g., some recent reviews).
REPLY: The reference was replaced by two recent reviews as below:
- Alegbeleye OO, Singleton I, Sant'Ana AS. Sources and contamination routes of microbial pathogens to fresh produce during field cultivation: A review. Food Microbiol. 2018;73:177-208.
- Iwu CD, Okoh AI. Preharvest Transmission Routes of Fresh Produce Associated Bacterial Pathogens with Outbreak Potentials: A Review. Int J Environ Res Public Health. 2019;16(22).
Comment 3. Line 58. The authors can cite reference 6 and add "as also shown in Lebanon (6)", the country of their interest.
REPLY: We added the requested expression “as also shown in Lebanon” in the Line 58.
Comment 4. Line 60. Ref 4. See comment above (line 54).
REPLY: The reference is replaced by a new recent review article as below:
- Alegbeleye OO, Singleton I, Sant'Ana AS. Sources and contamination routes of microbial pathogens to fresh produce during field cultivation: A review. Food Microbiol. 2018;73:177-208.
Comment 5. Lines 62-66. The specific reference is missing here and the data are not updated. The authors can update these data by looking at the list of multistate foodborne outbreak notifications provided by the CDC from 2006 to 2023 (https://www.cdc.gov/foodsafety/outbreaks/lists/outbreaks-list.html) and selecting the appropriate ones.
REPLY: By utilizing the National Outbreak Reporting System (NORS) provided by the CDC (https://wwwn.cdc.gov/norsdashboard/), we have successfully updated the data from 2006 to 2021. The latest information available on this website extends up to the year 2021. These data specifically pertain to multi-state outbreaks that were attributed to the consumption of fresh produce.
Page 2, Lines 63-66
These observations are confirmed by the analysis of 167 multistate outbreaks that occurred between 2006 and 2021 in the USA. The etiologic agents associated with these outbreaks were confirmed, and the source was largely attributed to cross-contamination within the distribution chain, poor agricultural practices, and fresh importations (10).
In addition, we added a new reference (Murray et al. 2017) related to microbial risk in the post-harvest of fresh produce to the expression “At postharvest, improper washing, handling, and storage among others have been implicated in produce contamination and associated outbreaks (9).”
- Murray K, Wu F, Shi J, Jun Xue S, Warriner K. Challenges in the microbiological food safety of fresh produce: Limitations of post-harvest washing and the need for alternative interventions. Food Quality and Safety. 2017;1(4):289-301.
Comment 6. Line 110: The reference to the climatic conditions of a Thai (ref. 24) seems inappropriate. Try to replace it with a more geographically appropriate one.
REPLY: The initial reference has been replaced by another appreciate one as below.
- Hazards EPanel oB, Koutsoumanis K, Allende A, Alvarez-Ordóñez A, Bolton D, Bover-Cid S, et al. Public health risks associated with food-borne parasites. EFSA Journal. 2018;16(12):e05495.
Comment 7. Material and Methods: The "sample collection criterion" is weak and the methodology used for vegetable processing and detection is not updated.
REPLY: We appreciate the valuable feedback offered by the reviewer, and we have duly acknowledged the limitations of our study, specifically the non-use of molecular diagnostic tools and the relatively low number of vegetable samples. These constraints were primarily due to the challenges posed by the COVID-19 pandemic and the ongoing Lebanese crisis, which led to logistical difficulties.
It is essential to note that our study encompassed a collection of 300 fresh vegetable samples from 68 different local markets in Tripoli (Lebanon), with an annual collection of 150 samples. While we acknowledge that the sample size might not be as extensive as studies with thousands of samples, we firmly believe that our research provides significant pilot data that address critical questions about the prevalence of parasitic contamination and related risk factors during the marketing phase of commonly consumed vegetables in North Lebanon.
Furthermore, it is important to highlight that this study was carried out by a team of highly experienced parasitologists with extensive expertise spanning many years.
By offering these preliminary findings, our study lays the groundwork for further investigations and paves the way for future research in this area. Despite the limitations, the data we present sheds light on important aspects of vegetable safety and can aid in implementing targeted interventions to mitigate risks associated with parasitic contamination in the region.
Comment 8. The paper lacks a "study design" that would have included a priori (and not ex-post) selection of collection sites, collection frequency and collection conditions.
REPLY: This is a pilot cross-sectional study. The study design has been well clarified in the M&M section.
We have made significant efforts to incorporate various expressions pertaining to the selection of collection sites, frequency of collection, and collection conditions in the manuscript. These additions aim to strengthen the study design and enhance the clarity and comprehensiveness of our research.
- We added at the end of the paragraph 2.1. “The local market selection was conducted using a random sampling approach to achieve a representative distribution across the Tripoli district, as depicted in Figure 1. To assess the microbial risk inherent in Lebanese traditional cuisine, five leafy green vegetables frequently consumed as raw materials were purposefully chosen, including mint, lettuce, parsley, arugula, and purslane”.
- We reformulate the content of the paragraph 2.2. to avoid having redundance of ideas and we added an expression about the frequency of collection: “total of 300 fresh vegetable samples were collected from 68 local markets in Tripoli over two consecutive years (2020-2021), with an annual collection of 150 samples. Fifteen samples of vegetable samples were collected weekly over a period of 10 weeks (from 1st of September till mid of November).”
Comment 9. In the absence of a robust true "prevalence study", the term "prevalence" used throughout the text cannot be accepted.
REPLY: Based on our study design, methodology, and findings, we provided a robust prevalence of parasitic contamination of commonly consumed vegetables in North Lebanon. Moreover, to enhance the transparency and clarity of our findings, we calculated the 95% confidence interval for the prevalence and included this information in both the abstract and the main text. By doing so, we aimed to provide a more comprehensive understanding of the potential variability in the prevalence estimates.
Comment 10. Lines 113-114: add the suggested words: "A total of 68 local markets located "in different areas" of the city of Tripoli".
REPLY: Done.
Comment 11. Line 135: add the weight of the original samples.
REPLY: We bought single bundle/batch of each leavy green vegetable. The expression turns into “Single batch/bundle of vegetable sample was purchased once from each seller”.
Comment 12. Line 141: Why is the sample weight in a range of 30-40 g?
REPLY: The sample weight varies widely in the literature from 250 mg of vegetable sample (Khan et al. 2022 Brazilian Journal of Biology, 2022, vol. 82, e242614), 25 g (Akoachere et al. 2018 BMC Res Notes 11:100) till one kilogram of vegetable (Yusof et al. 2017 Tropical Life Sciences Research, 28(1), 23–32).
In the Lebanese market, the leafy green vegetables are usually sold in bundle/bouquets/batches. The Lebanese daily salad prepared for 3-4 persons contains generally one bouquet (approx. 30g) of each of our selected vegetable. Therefore, we weighted 30g for Arugula, Mint, Parsley, and Purslane, as well as 40 g for Lettuce. All the roots were removed and only the green leaves were used.
Comment 13. Results: Line 184: The authors state that the variables considered (and also listed in Table 1) were not statistically significant. Please clarify in the M&M why they chose these variables.
REPLY: Done. We added a paragraph to explain the chosen variables.
Page 4, Lines 139-143: The open-air market is defined as an outdoor market that take place in an open public space. The closed market is an indoor market that operates within an enclosed or cov-ered space. All the above factors were investigated as potential risk factors for parasitic contamination in the retail settings.
Comment 14. Remove and/or replace the term "prevalence" with "presence"/"percentage of contamination" where appropriate (in the abstract, Figure 2; lines 198; 238; 276; 280).
REPLY: Thank you for raising this question. Based on our experience and the cross-sectional design of our study, we believe that the term “prevalence” is an appropriate outcome measure. In cross-sectional studies, such as the one we conducted, prevalence is commonly employed to assess the proportion of individuals within a population who have a particular disease, condition, behavior, or characteristic at a specific point in time.
Comment 15. Regarding the title, I would kindly suggest removing "high prevalence" and use a more “honest” title.
REPLY: We removed the “high prevalence” expression from the title and the new one is “Parasitic Contamination of Fresh Leafy Green Vegetables Sold in Northern Lebanon”.
Comment 16. Please use the term "sp. and/or spp." correctly throughout the text
REPLY: Done.
Comment 17. Discussion: As mentioned above, the authors state in the Results section that the variables considered (and also listed in Table 1) were not statistically significant. These results should be discussed in the Discussion section.
REPLY: We have discussed the non-significant variables ate the end of the discussion before the limitation of the study. The retail market factors examined in this study did not show a significant association with parasitic contamination, aligning with findings from prior research where variables such as vegetable display methods (58, 59), vegetable types (60) and market storing status (58) were also found to be non-significant in their impact. In addition. the wetness status of the vegetables analysed in this study are attributed to practices such as surface sprinkling with water. A study in Sudan revealed a high contamination of the water used to sprinkle vegetables with Strongyloides larvae, Giardia duodenalis, Entamoeba coli, and Ascaris lumbricoides (30). The lack of significance in our study may be attributed to the low number of wet vegetable samples. In this context, previous investigations in Turkey, Iran and Ethiopia showed that washing of vegetables contributes for significant decrease in the rate of parasitic contamination (58, 61, 62).
Comment 18. The authors point out some limitations of their contribution, including the lack of molecular tools, as microscopy itself is known to underestimate parasite burden. In this respect, it is highly recommended to cite the following paper: https://doi.org/10.1016/j.ijfoodmicro.2022.109634. Indeed, the authors could add the following sentence: "Molecular investigation allows to overcome significantly the limits of microscopy, as recently shown by Barlaam et al, 2022 in a study carried out in Italy on fresh produce" or something similar.
REPLY: We added the above cited papers in the line 306 “Molecular investigations, as shown by Barlaam et al. (2022) in their recent study conducted in Italy, have represented a valuable advancement compared to traditional microscopy, particularly in the analysis of fresh produce (65)”.
Comment 19. Line 240: As the authors know, Ascaris lumbricoides and Ascaris suum are now synonymous. Therefore, the reference to humans only in this sentence is a limitation.
REPLY: We removed A. lumbricoides and replaced it by Ascaris spp.
Comment 20. Lines 247-258: Please also refer to Giardia detection in school children by Osman et al, 2016 in Lebanon.
REPLY: We have referred to the article of Osman et al. 2016 in this part and we compared between the percentage of infection in humans and the percentage detected in vegetables. In the Lines 268-271, we added the following paragraph:
However, Giardia duodenales was detected among 25.8% of schoolchildren in Tripoli in 2016 (25) and only 1% of analyzed vegetable sample was positive. Giardia cysts were found to exhibit a limited ability to survive on fresh produce for an extended period, as 50% of the cysts died within the first 24 hours (53).
Comment 21. Reference 7: The link to this reference is missing.
REPLY: The link was added to reference 7.
Faour-Klingbeil D. The Microbiological Safety of Fresh Produce in Lebanon- A holistic “farm-to fork chain” approach to evaluate food safety, compliance levels and underlying risk factors. England: University of Plymouth; 2016 [Available from: https://core.ac.uk/download/pdf/80690128.pdf].
Comment 22. Figure 1 is not needed. It does not add any important information.
REPLY: In response to your previous comments, we have taken the opportunity to reorganize the Methods and Materials section to enhance clarity and coherence. As part of this improvement, Figure 1 allows illustrating the process of random selection of local markets for vegetable collection. The distribution of these markets spans across the entire city of Tripoli, aligning with the strategic study design we employed. We believe that the inclusion of Figure 1 strengthens the overall presentation of our study methodology.
Comment 23. Table 1: The meaning of superscript 1 should be explained in a footnote.
REPLY: Done. The meaning of the superscript 1 has been added in the footnote of the table 1. “1 Reference group”
Comment 24. Abstract - Line 28: Add "some" protozoa. For example, the technique used did not allow the detection of Cryptosporidium.
REPLY: Done. We added the word “some” before Protozoa in the line 23.
Comment 25. English language - Although I am not a native English speaker, I think the manuscript needs to be revised by a native speaker to remove some errors or inaccuracies.
REPLY: The paper has been checked by two native English speakers within Cornell University (USA) and University of Georgia (USA). The new identified errors are highlighted in yellow.
Reviewer 2 Report
The manuscript titled "High prevalence of parasitic contamination of fresh leafy green vegetables sold in northern Lebanon" aimed to investigate the foodborne parasitic contamination of vegetables sold in Tripoli public markets in northern Lebanon. Although the main findings of this study indicates a high incidence of parasitic elements in the sampled vegetables that can lead to foodborne diseases, further studies are needed to obtain more reliable information that could lead to improving public health and raising food safety awareness among the population of the studied area.
Some revisions are needed before publication in Pathogens.
-Lines 100-103: Please reformulate the objective of the study. I believe that due to some limitations of this study, the authors cannot say that the final result will provide important data for public health organizations. As mentioned earlier in the study by the authors, the methodology and sample size are important limitations in this case.
-It is not clear how many markets were involved in the study. Also, the total number of different vegetables collected from each market. I suggest that this data be included in Table 1.
- The authors should indicate in both the abstract and the M&M that the method used for parasite detection was qualitative microscopy.
- Please explain the terms "open air market" and "closed market" in more detail. If the authors are referring to the moisture status of the vegetables collected (wet/dry), were all samples in plastic bags or some without? Those that were wet, perhaps?
- Lines 148-149: Please add a reference.
- Authors should update the reference list including more recent publication on the topic, e.g.
- Barlaam A, Sannella AR, Ferrari N, Temesgen TT, Rinaldi L, Normanno G, Cacciò SM, Robertson LJ, Giangaspero A. Ready-to-eat salads and berry fruits purchased in Italy contaminated by Cryptosporidium spp., Giardia duodenalis, and Entamoeba histolytica. Int J Food Microbiol. 2022 Jun 2;370:109634. doi: 10.1016/j.ijfoodmicro.2022.109634.
do Nascimento Ramos IC, Ramos RAN, Giannelli A, Lima VFS, Cringoli G, Rinaldi L, de Carvalho GA, Alves LC. An Additional Asset for the FLOTAC Technique: Detection of Gastrointestinal Parasites in Vegetables. Acta Parasitol. 2019 Jun;64(2):423-425. doi: 10.2478/s11686-019-00059-3.
- I suggest adding some photos showing on the parasitic elements detected in the samples (Giardia, Entamoeba, Blastocystis, hookworms, etc.).
Moderate editing of English language required
Author Response
Dear Reviewer 2,
We are pleased that you found our manuscript interesting, and we thank you for the thoughtful reading and constructive comments.
Please find a revised version of our manuscript. As requested, we answered all your comments and suggestions. All answers are listed below and included in the revised manuscript.
Thank you for considering this revised version of our manuscript.
Comment 1. Lines 100-103: Please reformulate the objective of the study. I believe that due to some limitations of this study, the authors cannot say that the final result will provide important data for public health organizations. As mentioned earlier in the study by the authors, the methodology and sample size are important limitations in this case.
REPLY: We reformulated the second objective to be “The ultimate objective is to provide valuable insights that can be utilized in future research or in combination with other findings to inform public health and agricultural policies and practices”
Comment 2. It is not clear how many markets were involved in the study. Also, the total number of different vegetables collected from each market. I suggest that this data be included in Table 1.
REPLY: Done (Check Page 3, Lines 127-130). The number of markets is too high, so Table 1 will be complex if we want to add this information and the number of samples collected from each market as it is variable (from 1 till 5).
Comment 3. The authors should indicate in both the abstract and the M&M that the method used for parasite detection was qualitative microscopy.
REPLY: Done. The word Qualitative has been added to the abstract line 23 and in the M&M Line 160.
Comment 4. Please explain the terms "open air market" and "closed market" in more detail. If the authors are referring to the moisture status of the vegetables collected (wet/dry), were all samples in plastic bags or some without? Those that were wet, perhaps?
REPLY: We added a definition for the terms in the lines 139-143 “The open-air market is defined as an outdoor market that take place in an open public space. The closed market is an indoor market that operates within an enclosed or covered space”.
The vegetables were displayed on ground or in boxes without plastic bags. Each vegetable sample was picked and transported by a sterile dry plastic bag to the lab for analysis.
Comment 5. - Lines 148-149: Please add a reference.
REPLY: Done. We added the following reference (Soulsby E. Helminths, arthropods and protozoa of domesticated animals. London: Baillere Tindall; 1982.)
Comment 6. Authors should update the reference list including more recent publication on the topic, e.g., Barlaam A, Sannella AR, Ferrari N, Temesgen TT, Rinaldi L, Normanno G, Cacciò SM, Robertson LJ, Giangaspero A. Ready-to-eat salads and berry fruits purchased in Italy contaminated by Cryptosporidium spp., Giardia duodenalis, and Entamoeba histolytica. Int J Food Microbiol. 2022 Jun 2;370:109634. doi: 10.1016/j.ijfoodmicro.2022.109634.
do Nascimento Ramos IC, Ramos RAN, Giannelli A, Lima VFS, Cringoli G, Rinaldi L, de Carvalho GA, Alves LC. An Additional Asset for the FLOTAC Technique: Detection of Gastrointestinal Parasites in Vegetables. Acta Parasitol. 2019 Jun;64(2):423-425. doi: 10.2478/s11686-019-00059-3.
REPLY: Done. We added both cited references in the lines 304-308 (Page 8):
“Recently, the FLOTAC technique has shown an additional asset by detecting gastrointestinal parasites in vegetables in Brazil (64). Molecular investigations, as shown by Barlaam et al. (2022) in their recent study conducted in Italy, have represented a valuable advancement compared to traditional microscopy, particularly in the analysis of fresh produce (65).”
Comment 7. - I suggest adding some photos showing on the parasitic elements detected in the samples (Giardia, Entamoeba, Blastocystis, hookworms, etc.).
REPLY: We sincerely appreciate your suggestion to include photos of the parasites detected in the samples, to enrich the presentation of our findings. Indeed, visual representations can be highly beneficial in conveying scientific information effectively. Regrettably, due to logistic reasons and the constraints of our laboratory facilities in Lebanon, we do not have access to a microscope equipped with a camera, which limited our ability to capture and present high-resolution images of the identified parasitic elements.

Reviewer 3 Report
The manuscript entitled “High prevalence of parasitic contamination of fresh leafy green vegetables sold in northern Lebanon” point out the parasite contamination of routinely consumed vegetables. The authors collected samples from several markets spread all over Tripoli region through 2020 and 2021. Some weaknesses of this study were already mentioned by the authors in the text, and some other points are raised below:
1- Are there some missing references in the paragraph of lines 62-66.
2- Is there a difference of parasites found in each year? In the table 1 was shown the presence of parasites in general in 2020 and 2021, but it would be more elucidative if the sample types were distinguished by years too.
3- What is the significance of the data over the region where the samples were collected from? There is no information of intestinal parasites prevalence through the zones where the samples were collected. This information could bring data of more impacted region and maybe a track of vegetables supplier and facilitate the identification of possible contamination sources.
4- Despite the lack of a molecular approach to identify the parasites and the low number of collected samples, it would be more elucidative if the authors bring qualitative images of the parasite found in the samples.
5- The authors mentioned in the line 197 that this is the first study in this field, however in the line below it was mentioned a study performed in Bekaa Valley that corroborates the data within this manuscript. Independently of the chosen region, this was definitively not the first study in Lebanon. Moreover, what is the relation of the relation of Bekaa Valley and Tripoli region? Is it the same river that runs through both regions? Are the vegetables distributed in the markets of both regions related in terms of planting locations? This information would be very valuable to understand the discussions related to the cited work.
Author Response
Dear Reviewer 3,
We are pleased that you found our manuscript interesting, and we thank you for the thoughtful reading and constructive comments.
Please find a revised version of our manuscript. As requested, we answered all your comments and suggestions. All answers are listed below and included in the revised manuscript.
Thank you for considering this revised version of our manuscript.
Comment 1. Are there some missing references in the paragraph of lines 62-66.
REPLY: Done. We have added a reference to the mentioned expression “Between 2014–2021 (…) underreported (3)
3. CDC. Lettuce, Other Leafy Greens, and Food Safety 2023 [Available from: https://www.cdc.gov/foodsafety/communication/leafy-greens.html.
Comment 2.- Is there a difference of parasites found in each year? In the table 1 was shown the presence of parasites in general in 2020 and 2021, but it would be more elucidative if the sample types were distinguished by years too.
REPLY: Thank you for your interest and observation regarding the presence of parasites in different years of our study. This analysis was already done in our manuscript using multivariable logistic regression models. After carefully analyzing the data, we found no significant difference in the prevalence of parasites after accounting for date, sample type, market storing status, and purchase status. This rigorous statistical approach has allowed us to account for potential confounding factors, ensuring the validity and reliability of our results.
Comment 3. What is the significance of the data over the region where the samples were collected from? There is no information of intestinal parasites prevalence through the zones where the samples were collected. This information could bring data of more impacted region and maybe a track of vegetables supplier and facilitate the identification of possible contamination sources.
REPLY: Several epidemiological studies have been conducted to study the prevalence of the protozoan parasitic infections in the North of Lebanon. The comparison between the data obtained in vegetables and those previously identified among Lebanese community living in North Lebanon was presented in the Lines 265-276 (Page 7). We have added the expression “in Northern Lebanon” in line 267 to make it clearer for the readers.
Comment 4. Despite the lack of a molecular approach to identify the parasites and the low number of collected samples, it would be more elucidative if the authors bring qualitative images of the parasite found in the samples.
REPLY: We sincerely appreciate your suggestion to include photos of the parasites detected in the samples, to enrich the presentation of our findings. Indeed, visual representations can be highly beneficial in conveying scientific information effectively. Regrettably, due to logistic reasons and the constraints of our laboratory facilities in Lebanon, we do not have access to a microscope equipped with a camera, which limited our ability to capture and present high-resolution images of the identified parasitic elements.
Comment 5. The authors mentioned in the line 197 that this is the first study in this field, however in the line below it was mentioned a study performed in Bekaa Valley that corroborates the data within this manuscript. Independently of the chosen region, this was definitively not the first study in Lebanon.
REPLY: This is the first study about the parasitic contamination of vegetables sold in Lebanese markets.
Abi Saab et al (2022) have evaluated the potential health risks of irrigating vegetables (radishes, parsley, onions, and lettuce) using three water sources (groundwater, river water, and treated wastewater) and three irrigation methods (drip, sprinkler, and surface) over two growing seasons in 2019 and 2020. Water, crop, and soil samples were analyzed for physicochemical parameters, pathogens, and metals. The vegetables tested in this study were not sold in the markets and they were harvested directly from the soil after been irrigated by different types of water.
We reformulated the context in the manuscript to clarify the difference between both studies.
Our results highlighted a relatively high prevalence of parasitic contamination in vegetables sold in North Lebanon, which aligned with a previous study reporting a prevalence of 16.67% in vegetables directly harvested from the Bekaa Valley in 2020 (34). The previous study suggested that the water used for irrigation, sourced from the Litani river, had significant levels of contamination with Escherichia coli and some parasite species, surpassing the threshold limits recommended by the FAO (35), in comparison to groundwater and treated wastewater. Moreover, crops irrigated with this water showed the highest percentage of parasitic contamination when compared to those irrigated with groundwater and treated wastewater.
Comment 6. Moreover, what is the relation of the relation of Bekaa Valley and Tripoli region? Is it the same river that runs through both regions? Are the vegetables distributed in the markets of both regions related in terms of planting locations? This information would be very valuable to understand the discussions related to the cited work.
REPLY: The Bekaa Valley is a distinct geographical region located in eastern Lebanon. It is a fertile and agricultural region, counting for 37% of total production of vegetables in Lebanon. Indeed, the Litani River flows through the Bekaa Valley and has been a major source of water for irrigation in the region. It does not directly run through the Tripoli region.
Tripoli is a coastal city and region located in the northern part of Lebanon. It is the second-largest city in the country and has its own local geography and river systems. The Kadisha-Abou Ali River is relevant to the Tripoli region and has already been demonstrated that it is highly contaminated by wastewater, like the Litani River.
The vegetables distributed in the markets were sourced from different farms located in Bekaa and North Lebanon. This has been mentioned in the lines 138 – 142 (Page 4).
We also added the location of the Bekaa valley in Eastern Lebanon by comparison to Tripoli in North Lebanon.
“The vegetables were sourced from different farms and agricultural areas located in both the Bekaa Valley, a region in Eastern Lebanon that accounts for approximately 37% of the country's vegetable production (29), and North Lebanon.”

Round 2
Reviewer 3 Report
Now the manuscript is ready for publication.
Author Response
Thank you again for your constructive comments and suggestions.